# Implication of the LRR Domain in the Regulation and Activation of the NLRP3 Inflammasome

**DOI:** 10.3390/cells13161365

**Published:** 2024-08-16

**Authors:** Margaux Cescato, Yixiang Y J Zhu, Laurent Le Corre, Bénédicte F Py, Sophie Georgin-Lavialle, Mathieu P Rodero

**Affiliations:** 1Laboratory of Pharmacological and Toxicological Chemistry and Biochemistry, CNRS, Paris Cité University, 75006 Paris, France; margaux.cescato@gmail.com (M.C.); zhu.yvesjean@gmail.com (Y.Y.J.Z.); laurent.le-corre@parisdescartes.fr (L.L.C.); 2National Reference Center for Autoinflammatory Diseases and AA Amyloidosis, Department of Internal Medicine, Tenon Hospital, Sorbonne University, Assistance Publique—Hôpitaux de Paris (APHP), 75020 Paris, France; sophie.georgin-lavialle@aphp.fr; 3CIRI, International Center for Research in Infectiology, Inserm, University Claude Bernard Lyon 1, 69007 Lyon, France; benedicte.py@ens-lyon.fr

**Keywords:** inflammasome, NLRP3, Leucine-Rich Repeat domain, LRR, CAPS

## Abstract

The NLRP3 inflammasome is a critical component of the innate immune response. NLRP3 activation is a tightly controlled process involving an initial priming to express NLRP3, pro-IL-1 β, and pro-IL-18, followed by an activation signal. The precise mechanism of activation is not fully understood due to the diverse range of activators, yet it effectively orchestrates the activation of caspase-1, which subsequently triggers the release of proinflammatory cytokines IL-1β and IL-18. NLRP3 dysregulation can lead to a variety of inflammatory diseases, highlighting its significant role in immune response and disease pathogenesis. NLRP3 is divided into three domains: the PYD, the NACHT, and the LRR domains. This review focuses on the LRR domain of NLRP3, detailing its structural characteristics, its function in pathogen sensing, its role in the degradation process, and its involvement in inflammasome auto-inhibition and activation. Additionally, we discuss the impact of mutations within the LRR domain found in atypical Cryopyrin-Associated Periodic Syndromes (CAPS), highlighting the clinical relevance of this domain.

## 1. Introduction

Innate immune cells recognize pathogens or danger-associated molecular patterns (PAMPs and DAMPs) through pattern-recognition receptors (PRRs) such as Toll-like receptors (TLRs), C-type lectin receptors (CLRs), RIG-I-like receptors (RLRs) and NOD-like receptors (NLRs) [1].

NLR proteins are mostly cytosolic receptors that can sense a wide range of microbial molecules [2,3], leading to either the induction of an inflammatory response or the regulation and/or inhibition of inflammation [4]. Additionally, NLRs have been found to support immune homeostasis in tissues [5]. There are 23 NLRs described in humans and 33 in mice [6]. They all share common features, including (i) an N-terminal effector domain; (ii) a central nucleotide-binding domain termed NACHT (NAIP, CIITA, HETE, and TP1) or NOD, which is involved in ATP-dependent oligomerization; and (iii) a C-terminal portion consisting of a repetition of Leucine-Rich Repeats (LRRs; Figure 1). NLRs can be classified into subfamilies based on their N-terminal effector domain: the NLRA and NLRC families have acidic transactivation and caspase recruitment (CARD) domains, the NLRB family possesses a baculoviral inhibitory repeat (BIR)-like domain, the NLRP family contains a pyrin domain (PYD) and the NLRX family is defined by a mitochondrial targeting sequence (MTS) [7].

NLRP3, previously “CIAS1” or cryopyrin, is a member of the NLRP protein family. Its activation leads to the formation of the inflammasome complex [16], which are multi-protein complexes that, once assembled in an activated conformation, recruit and activate caspase-1. Activated caspase-1 subsequently cleaves various substrates, resulting in the release of proinflammatory cytokines such as IL-1β and IL-18 and the cleavage of gasdermin-D (GSDMD) into fragments that disrupt the membrane by forming pore, leading to pyroptotic cell death and facilitating cytokine release [17,18]. 

NLRP3 activation is tightly regulated, and three activation mechanisms have been identified. The canonical activation requires both priming and activation steps (Figure 2). The priming signal can be referred to as a PAMP. It is mediated through activation of PRRs [10] or cytokine receptors [19,20,21]. The priming process serves two main purposes: first, it upregulates the expression of inflammasome components such as NLRP3, pro-IL-1β, and pro-IL-18 [19] via signaling pathway involving MYD88, TRIF, and NF-kB [22,23], and second, it leads to post-translational modifications (PTMs) of NLRP3 [24,25]. Following the activation signal, NLRP3 forms the inflammasome complex with NEK7, ASC, and caspase-1. NLRP3 is unique in its broad range of activators, which include pathogens components [26,27,28,29,30,31,32,33,34], endogenous signals [35,36,37,38,39,40,41,42,43,44,45], and crystals [46,47,48]. However, the exact mechanisms by which these various stimuli activate NLRP3 remain unknown. While it is proposed that these signals converge on potassium (K+) efflux [36], studies have also shown K+ efflux-independent NLRP3 activation [49], suggesting a complex regulatory network involving both K+ efflux-dependent and independent pathways.

An alternative activation pathway has been described in human monocytes treated with LPS (Figure 2). In this context, the TRIF pathway controls the assembly of an alternative NLRP3 inflammasome that does not form the typical ASC speck and which triggers IL-1β release independently of cell death in a gradual manner [50]. NLRP3-dependent IL-1β release by living cells, termed hyperactivation as it results in a high amount of IL-1β over time, has also been reported in mouse cells treated with oxidized lipids [51].

In the non-canonical pathway (Figure 2), cytoplasmic LPS, independently of TLR signaling, is sufficient to form the inflammasome [52]. LPS directly binds and activates caspase-4 and 5, leading to gasdermin cleavage and pyroptosis [53]. Pores in the membrane cause K+ efflux, triggering NLRP3 inflammasome formation [54].

NLRP3 activation is intricately regulated due to its pivotal role in cellular homeostasis, and dysregulation in this process can lead to pathological conditions. The NLRP3 inflammasome is implicated in several frequent inflammatory diseases, including gout [55,56], rheumatoid arthritis [57], Crohn’s disease [58], and, more occasionally, pulmonary fibrosis [46].

Metabolic stressors such as ceramide and fatty acids accumulation, commonly observed in obesity, can also trigger NLRP3 activation, contributing to establishing an inflammatory landscape that promotes insulin resistance and facilitates the onset of type 2 diabetes [44,45]. Additionally, NLRP3 activation can also be triggered by oxidized low-density lipoprotein, leading to atherosclerosis [43]. NLRP3 is also proposed to be activated in several neurodegenerative disorders, including Alzheimer’s disease [40,41,42], Parkinson’s disease [59,60] and epilepsy [61,62]. Furthermore, gain-of-function heterozygous variants in *NLRP3* are associated with a rare monogenic auto-inflammatory disease called Cryopyrin-Associated Periodic Syndrome (CAPS) and characterized by systemic inflammation and various clinical features, including mainly urticaria, arthromyalgia, buccal aphthous, and neuroinflammation with eye, hearing, and meningitis involvement [63].

There are two different annotation systems for NLRP3, which may be confusing in the literature as the first exon has two potential start codons [64]. In one system, NLRP3 is annotated as having 1036 amino acids (UniProt), starting the count from the first ATG. However, in the other system, the count begins from the second ATG, which has a better Kozak consensus, resulting in a length of 1 034 amino acids. Furthermore, the first methionine residue is not present in the murine form [65]. This discrepancy of 2 amino acids can result in variations across different publications. For instance, a well-known mutation in NLRP3 is either called the R260W [66] or the R262W mutation [67]. Furthermore, NLRP3 is conserved across species, but there are slight differences between the human and murine forms. Murine NLRP3 consists of 1 033 amino acids and shares 89.6% similarity and 82.7% identity with its human counterpart [68]. This disparity is significant, leading to activation differences. For instance, LPS alone can activate NLRP3 in human cells through an alternative activation pathway but not in murine cells [50,69]. Additionally, while low doses of dynasore, a dynamin inhibitor, can activate NLRP3 in human cells, a 16-fold higher dose is required to activate murine NLRP3 [70]. These findings suggest that human cells are overall more sensitive to inflammasome activation than murine cells.

NLRP3 is divided into three domains: the PYD domain, responsible for interactions with downstream proteins [71] of the inflammasome complex; the NACHT domain containing ATPase activity; and the LRR domain. This review aims to provide a comprehensive overview of current knowledge and recent discoveries regarding the involvement of the LRR domain in modulating NLRP3 inflammasome activation.

## 2. Structure of NLRP3 LRR Domain

There are two recognized versions of the NLRP3 gene nomenclature: one version includes 10 exons [72], while the other comprises a total of 9 exons [73,74]. In the latter version, 6 of these exons are forming the LRR domain (Figure 3). Except for exon 9, which encodes only one LRR, each exon within the LRR domain has a length of 171 base pairs and encodes 2 LRRs. Consequently, the LRR domain comprises a repetitive sequence of 11 LRR patterns, each consisting of either 28 or 29 amino acids, characterized by a high leucine content and a consensus sequence of xLxxLxLxxN/CxLxxxxxxxLxxxLxxxxx [72] (Figure 4A,B). This repetitive pattern forms a uniform and curved horseshoe-like structure with parallel beta sheets on the concave side and α-helical elements on the convex side (Figure 4A,C in blue) [72]. The precise alternating arrangement of LRRs and the exon-exon border always positioned at the same position facilitates the generation of functional isoforms through alternative splicing [72], as the exon-exon junction and the reading frame are conserved without disrupting the 3D structure. Beyond its high leucine content, NLRP3 contains a significant amount of redox-sensitive cysteine (4.3% in human NLRP3 and 4.6% in murine NLRP3), compared to average NLR proteins at around 2.3%. These conserved cysteine residues are located in the LRR domain (Figure 4A,C in red), suggesting that the LRR domain may function as a sensor for ROS [75].

In 2002, three different NLRP3 isoforms were initially identified through reverse transcription and specific amplification of the NLRP3 gene from human blood. Sequencing revealed that the most prevalent isoform was the long isoform containing all nine exons. Additionally, shorter and intermediate isoforms were identified, with the shorter variant lacking exons 4 to 9, resulting in a loss of the LRR domain, and the intermediate variant lacking exons 4 and 6 [73] (Figure 5).

Subsequent RNAseq analysis on human monocyte-derived macrophages from healthy donors confirmed the predominance of the full-length isoform and uncovered additional isoforms. Exon 4 was found to be the most frequently skipped exon, and therefore, NLRP3Δ4 emerged as the second most abundant variant after the full-length. Other variants, such as NLRP3Δ6 isoforms and NLRP6Δ4/6, were identified as well [72].

Further analysis of archive data from experiments using human primary monocytes revealed a total of 14 NLRP3 isoforms. Among them, full-length NLRP3 and NLRP3Δ4 isoforms remained the most prevalent [74]. While full-length NLRP3 is an active form, NLRP3Δ4 is inactive supposedly due to its inability to bind NEK7 [72], resulting in the incapacity to recruit ASC, assemble an inflammasome and the absence of IL-1β production [72,74]. Similarly, NLRP3Δ5, NLRP3Δ7, and NLRP3Δ9 have lost their competence to form an active inflammasome. In contrast, NLRP3Δ6 and NLRP3Δ8 retain their ability to form the inflammasome and produce IL-1β [74]. This observation highlights that not all LRRs are equally important. While exons 6 and 8 are dispensable for inflammasome activation, exons 4, 5, 7, and 9 are essential. This suggests that the specific amino acid composition within these exons plays a crucial role in enabling NLRP3 activation. Given the observed loss of interaction with downstream proteins such as NEK7, LRR is required to form a functional inflammasome that nucleates ASC filaments.

Interestingly, even gain-of-function variants in the NACHT domain responsible for CAPS, such as R260W, do not rescue the activity of non-active isoforms, reinforcing the necessity of these LRR exons to form a functional inflammasome. Importantly, the activity of full-length NLRP3 was not affected by the co-expression of non-functional isoforms, suggesting they do not act as dominant negative nor impair the activation of full-length NLRP3 [74]. Finally, LPS stimulation increased the expression of the full-length NLRP3 isoform, indicating that the repertoire of NLRP3 isoforms can be modulated by exogenous signals [74].

## 3. Pathogen Sensing

Pieces of evidence indicate that pathogen recognition in various species is mediated by LRR domains. In plants, immune proteins encoded by R-genes feature LRR domains. Sequence analysis revealed that their specificity to pathogens signals depend on these domains [77]. In mammals, TLRs also possess LRR domains with a characteristic 3D structure similar to that of NLRs. They bind ligands as extracellular domains on their concave surface [78].

Similarly to TLRs, NLR responses are highly agonist-specific. Within the NLRC subfamily, NOD1 and NOD2 not forming inflammasome can detect bacterial components to activate NF-kB. Early research demonstrated the crucial role of LRR domains in bacterial sensing for NOD1 and NOD2 [79]. Truncation of the LRR in NOD2 led to mutants unable to respond to bacterial components, and similarly, NOD1 lacking LRR lost its responsiveness to LPS [80]. This suggests that LRR domains in NLRC proteins play an essential role in recognition, highlighting the critical importance of LRR integrity [79].

In the NLRP subfamily, NLRP1 responds to several bacteria, including *Shigella flexneri*, *Listeria monocytogenes*, and *Bacillus anthracis,* to form inflammasome [81,82] but also detect dsRNA through the direct involvement of LRR [83] supporting the detection role of LRR among NLRP proteins.

However, the role of the LRR domain of NLRP3 in DAMP sensing might not be as straightforward. Indeed, NLRP3 inflammasome is activated by a wide array of structurally and chemically diverse triggers. In addition to fungal [26,27,28,29], bacterial [30,31,32,32,33,34] and viral recognition [34], NLRP3 also responds to endogenous signals such as ATP [35], ROS [37], K+ efflux [36], crystals of monosodium urate (MSU) [55], extracellular matrix hyaluronan [39], amyloid-b [40,41,42], oxidized low-density lipoprotein [43], fatty acids [44,45] and oxidized mitochondrial DNA [38]. Furthermore, NLRP3 can sense environmental triggers like asbestos and silica [46,47,48].

Experiments have demonstrated that the LRR domain is dispensable for the response to nigericin, SiO_2_, alum, and *F. novidia* U112 in murine [84] and human cells [85]. However, the response to *L. monocytogenes* is impaired when the LRR domain is absent. Additionally, replacing the LRR domain of NLRP3 with that of NLRP2 results in a 50% reduction in activation by H_2_O_2_, suggesting the LRR domain of NLRP3 might possess features to recognize H_2_O_2_ [85]. Moreover, NLRP3, lacking half of its LRR domain, lost its ability to respond to MSU, a common trigger used to mimic gout [85].

Rather than direct activation, it appears that the diverse and highly various triggers for NLRP3 activation converge on cellular events such as cytosolic potassium (K+) efflux, production of ROS, or mitochondrial damage. Because these cellular events are interconnected and can mutually activate each other, identifying the causative event of NLRP3 activation is challenging. Several NLRP3 triggers, such as nigericin [86], ATP [87,88], and maitotoxin [32], induce membrane permeation, leading to K+ efflux. The physiological intracellular concentration of K+ has been shown to be too high for active NLRP3 conformation, and inhibition of K+ efflux prevents NLRP3 activation in both murine and human macrophages [89]. Therefore, it was suggested that cytosolic K+ efflux may be the critical common trigger for NLRP3 activation [36]. For K+ efflux-dependent stimuli, the use of ROS scavengers has shown no effect on inflammasome activation, suggesting that ROS are not directly implicated [36]. However, imiquimod, a small molecule known to be a ligand of TLR7, activates NLRP3 in a K+-independent but ROS-dependent manner, suggesting at least two independent pathways for NLRP3 activation [49]. Recent observations reconcile these two models by the demonstration that NLRP3 interaction with vesicles was critical for its activation regardless of K+ involvement. K+-dependent and independent NLRP3 activators increase the levels of phosphatidylinositol 4-phosphate (PI4P) in membranes, and this increase facilitates the binding of NLRP3 to PI4P, which is crucial for its activation [90,91,92,93].

## 4. Degradation

NLRP3 is degraded by both autophagy- and proteasome-dependant mechanisms [94,95] (Figure 6). Accumulative studies have highlighted the importance of PTMs within the LRR domain in the degradation of NLRP3. LPS priming has been shown to influence the half-life of NLRP3, extending it from 4 h to over 6 h [25]. This suggests that LPS alters the PTM status of NLRP3 to switch to a “ready-to-be-activated” conformation and increase its cellular level by reducing its degradation.

Studies have demonstrated that, in a resting state, NLRP3 bears degradative and non-degradative polyubiquitination with K63 and K48 ubiquitin chains [24,96] that keep NLRP3 both inactive and at low levels [94]. Therefore, upon activation, NLRP3 undergoes deubiquitination [24,97]. The LRR domain, rich in lysine residues, is a favorable site for ubiquitination, suggesting it might lead to NLRP3 degradation. However, intracellular levels of human NLRP3 variants lacking exons in the LRR domain [72,73,74] and murine NLRP3 mutants lacking LRR parts [84] are consistently lower compared to WT NLRP3. Moreover, in an overexpression system, NLRP3 lacking exons in the LRR domain are more prone to proteasomal degradation than the full-length NLRP3. This suggests that the C-terminal LRR domain plays a protective role, helping to prevent NLRP3 degradation [84].

Phosphorylation also participates in maintaining NLRP3 levels. For example, Y859 (Y861 in the other annotation) is phosphorylated under steady-state conditions. This phosphorylation is removed during the activation step by the phosphatase PTPN22 [98]. In mice, cells lacking PTPN22 show an increased proportion of NLRP3 in autophagosomes, suggesting that phosphorylation at Y859 induces NLRP3 degradation. When the tyrosine residue is mutated to a non-phosphorylable phenylalanine, the accumulation of NLRP3 in the autophagosome is prevented, further supporting the involvement of Y859 phosphorylation in the regulation of NLRP3 degradation. Specifically, SQSTM1, an autophagosome cargo protein, binds to phosphorylated NLRP3, facilitating its degradation through autophagy [99]. However, a fraction of NLRP3 is phosphorylated under steady-state conditions but is not recruited to phagophores, suggesting that multiple signals are involved in NLRP3 degradation, and Y859 phosphorylation alone is not sufficient. Moreover, the interaction between NLRP3 and SQSTM1 also depends on the pyrin domain [99], indicating that degradation is influenced by the accumulation of multiple signals.

Inflammasome activators also trigger PTMs controlling NLRP3 degradation, a mechanism that participates in its negative control and/or the resolution of the inflammation. The kinase Lyn phosphorylates NLRP3 following ATP activation at position Y916 (Y918 in the other annotation). This phosphorylation increases NLRP3 ubiquitination, leading to its degradation by proteasome, suggesting Lyn phosphorylation starts a process of ubiquitination, which leads to NLRP3 degradation [100]. Indeed, E3 ubiquitin ligase and deubiquitinase target sequentially NLRP3 following activation, determining its fate. For example, RNF125 adds K63 ubiquitin chains on the LRR domain, which controls the recruitment of Cbl-b, which in turn adds K48 ubiquitination chains on the NACHT domain targeting NLRP3 to degradation and represses inflammasome activation [101]. Similarly, the E3 ubiquitin ligase MARCH7 is implicated in the autophagy-mediated degradation of NLRP3, associated with K48-polyubiquitination within the LRR domain [102]. On the opposite, the recruitment of the BRCC36 deubiquitinase (as well as its mouse homolog BRCC3) by the ubiquitinated-LRR domain mediates its K63-deubiquitination following activation stimuli and promotes inflammasome assembly [96,103,104]. Noteworthy, K63 polyubiquitination of the LRR could also directly facilitate NLRP3 delivery to autophagosomes [105]. Indeed, NLRP3 interaction with CCDC50, an autophagy receptor highly expressed in immune cells [106], is significantly increased following LPS treatment. Although the specific binding motif has not been identified, CCDC50 triggers the degradation of K63-ubiquitinated NLRP3, with the LRR domain proving essential and sufficient for CCDC50 binding [105]. In addition to phosphorylation and ubiquitination, palmitoylation also plays a crucial role in regulating NLRP3 stability and degradation. Specifically, palmitoylation of the LRR domain at the cysteine C842 (C844 in the other annotation) by ZDHHC12 upon activation enhances the ability of the Heat Shock Cognate protein 70 (HSC70) to recognize NLRP3, which in turn promotes its degradation through the lysosomal pathway [107].

## 5. Activation/Auto-Inhibition

In the context of NLRP3 regulation, addressing auto-inhibition alongside activation is crucial for understanding the functions of the LRR domain.

The auto-inhibitory role of LRR domains is highly documented in various proteins. In plants, LRR domains are proposed to self-inhibit NLRs until pathogen sensing triggers its cleavage [108]. Similarly, the crystal structure of murine NLRC4 suggests that the LRR domain maintains NLRC4 in a monomeric inactive form [109]. Experimental evidence supports this observation, as NLRC4ΔLRR is constitutively active and oligomerizes spontaneously, regardless of flagellin presence [110,111]. Similarly, NLRP1ΔLRR binds ATP equally in the presence or absence of stimuli, enabling oligomerization in the absence of agonists [112]. Consistent with these findings, deletion of LRR domains in both NOD1 and NOD2 results in constitutive activation [79].

NLRP10 is the only member of the NLR family lacking a LRR domain (Figure 1) [113]. It was long thought to inhibit the NLRP3 inflammasome [114], but NLRP10 has recently been shown to form an inflammasome in response to mitochondrial damage in several murine and human models [14,115]. Both the PYD and NACHT domains are necessary for NLRP10 inflammasome formation, and the absence of the LRR domain does not prevent ASC recruitment [115].

Cryo-EM analysis of both human and murine NLRP3 reveals that inactive NLRP3 (bound to ADP/MCC950) forms various spheric oligomeric structures, including barrel-hexamer [116], decamer [76], dodecamer [117], and even tetradecamer and hexadecamer configurations [118] (Figure 7A). These NLRP3 “cages” sequestered the PYD domain within the complex core inaccessible to ASC and are maintained by LRR domains with interaction surfaces occurring in both “back-to-back” and “face-to-face” orientations (Figure 7A, left and middle). Specifically, the negatively charged C-terminal of one LRR domain interacts with the positively charged concave surface of another LRR [117], leading to the intertwining of LRR domains [76]. Multiple cage structures observed in cryo-EM suggest that inactive NLRP3 undergoes oligomerization, with LRR domains playing a crucial role in maintaining the overall structure. Further supporting the importance of LRR domains, an acidic loop (residues 686–725) acting in the LRR-LRR concave interface in the closed conformation becomes disordered in the active form [119]. Consistent with this hypothesis, NLRP3 mutants lacking the LRR domain show reduced self-oligomerization [120], and specific mutations in the “back-to-back” or “face-to-face” interfaces results in impaired cage formation with a higher quantity of free NLRP3 [118]. Notably, NLRP3 oligomers containing fewer NLRP3 units, such as hexameric cages, present a “head-to-face” surface interaction between a NACHT and LRR domain rather than the typical “face-to-face” surface between 2 LRR domains (Figure 7A, right). Several PTMs of NLRP3 upon priming may participate in the disruption of these auto-inhibited complexes, including phosphorylation of the LRR domain at S804 by CSNK1A1, and Y859, as only unphosphorylated forms are found in cage complexes [99,104,118].

Activated NLRP3 adopts a radial flower-like oligomeric structure, where LRR domains are not engaged in disk assembly. Instead, LRR domains bind NEK7 and lose their interactions with each other [119]. Notably, they do not interact with central caspase-1, which suggests LRR domains are not directly engaged in the catalytic part of the inflammasome, but rather, the binding to NEK7 potentially promotes the transition of NLRP3 from a closed to an open conformation controlled by ATP binding or stabilizes the latter (Figure 7B, left). Therefore, the role of LRR domains appears to precede inflammasome activation. Indeed, NLRP3 lacking the LRR domain does not exhibit constitutive activation under basal conditions [85]. The recent description of open NLRP3 oligomers in the presence of ATP revealed an octameric structure with conserved “back-to-back” and “face-to-face” interface interactions of the LRR domains (Figure 7B, middle). Additionally, new interface interactions termed “tail-to-tail” and “head-to-face” were identified by cryo-EM. This structure revealed a novel organization of NLRP3 oligomers proposed to be an intermediate form between inactive and active forms (Figure 7B, middle). Notably, whether in the open or closed conformation, NEK7 clashes with adjacent NLRP3, indicating that NEK7 presence leads to the dissociation of the oligomeric cage. This results in dimers of NLRP3/NEK7 with “back-to-back” interactions maintained through the LRR domains (Figure 7B, right) [116]. Consistent with earlier findings [119], the acidic loop located on the concave face of the LRR, which facilitates “face-to-face” interactions, is disordered in the open conformation [116]. However, it’s important to note that this recent model lacks the PYD domain, which could potentially influence the overall structure of the activated inflammasome.

NEK7, a key partner in inflammasome activation, is essential to NLRP3 inflammasome assembly in mouse macrophages [121,122]. Deletion experiments with ΔNEK7 cells have demonstrated altered caspase-1 cleavage [123] and impaired IL-1β production [124]. Co-immunoprecipitation assays and cryo-EM analysis have revealed that NEK7 binding depends on both the NACHT and LRR domains, although the precise involvement of the NACHT domain is unclear [123]. Notably, an intact LRR domain is essential for this interaction [120,121]. Supporting the concept of “cage opening,” the NEK7-NLRP3 interaction is moderately enhanced after LPS priming and significantly boosted following ATP activation [121], suggesting increased interaction between NEK7 and the LRR domain upon NLRP3 activation. The authors proposed that NEK7 may be hindered from interacting with the LRR domain under steady-state conditions due to neighboring LRR occupancy and auto-inhibition [119].

Further proving the significance of the LRR domain in the activation of the inflammasome, some LRR mutations have been proposed to either enhance or reduce the affinity for NEK7, rendering them pathogenic, such as R918Q and G755R, or hypomorphic, such as D946G, respectively [122,125]. In addition, NEK7 recruitment is also controlled by PTMs in the LRR domain. Following its transient phosphorylation upon priming signal, NLRP3 is dephosphorylated at S804 upon activation by nigericin. Phospho-mimetic substitution S804D prevents the recruitment of NEK7 and the inflammasome formation [104] (Figure 6A). The role of NLRP3 dephosphorylation at Y859 by PTPN22 on NEK7 recruitment remains to be investigated but could constitute an additional mechanism for the positive role of PTPN22 on inflammasome activity (Figure 6B). Indeed, the location of Y859 and S804 within the LRR domain, at the interface with NEK7, suggests a potential mechanism where steric hindrance or charge repulsion could inhibit binding between LRR and NEK7 [104,123]. Consequently, PTMs enhancing the affinity of the LRR domain for NEK7 could destabilize the inactive NLRP3 oligomers through interactions with the acidic loop, thereby promoting activation. Additionally, palmitoylation within the LRR domain was also shown to have an impact on NEK7 binding. Specifically, palmitoylations at positions C835 and C836 (C837/C838 in the other annotation) by ZDHHC5 promote NLRP3 activation following both priming and boost signals (Figure 6A,B). Modeling analysis suggests that palmitoylations at these sites may enhance the interaction between NEK7 and the LRR domain, facilitating the activation process [126]. The variant R918Q has been modeled to exhibit increased affinity for NEK7 [125]. This variant displays higher levels of palmitoylations on C835/C836, attributed to decreased recruitment of the Acylprotein thioesterase 1 and alpha/beta hydrolase domain-containing protein 17A depalmitoylase (ABHD17A). This reduced recruitment enhances palmitoylation and consequently strengthens binding to NEK7 [126].

Intriguingly, inactive NLRP3 isoforms lacking exon 4 lose the ability to bind NEK7 [72], indicating the critical importance of amino acids other than Y859 (in exon 6), S804 (in exon 5) or R918 (in exon 7) for NEK7 binding, or the necessity of an intact LRR domain for NEK7 interaction [120]. However, some studies suggest that the LRR domain may not be necessary for interaction with NEK7 and subsequent inflammasome activation [84,85] and that NLRP3 can also form the inflammasome independently of NEK7 [23,127]. Notably, long priming that activates IKKβ has been reported to enable NEK7-independent inflammasome activation as the predominant pathway in human myeloid cells [23]. Further studies would be required to investigate the putative differential role of the LRR domain upon the IKKβ-dependent vs NEK7-dependent inflammasome activation pathway.

In addition to NEK7 recruitment, the LRR domain also facilitates the recruitment of other partners upon activation, such as the AKT kinase. During the priming phase, AKT phosphorylates serine 5 in the PYD domain, preventing NLRP3 from interacting with the adaptor protein ASC [128]. Furthermore, the SGT1-HSP90 (co-chaperon-like ubiquitin-ligase-associated protein and heat-shock protein 90) complex is also recruited by the LRR domain to aid inflammasome activation [129], and disruption of this interaction inhibits NLRP3 activation [130]. Furthermore, HSP90β and SGT1 are essential for the self-activation of NLRP3 variants that carry activating mutations, further proving the involvement of the LRR domain in recruiting activating partners [131] (Figure 6B). Interestingly, SGT1 interacts minimally with non-active NLRP3 isoforms but interacts with full-length NLRP3 and isoforms lacking exon 6 or 8, which are the only active isoforms [74]. It suggests that the LRR domain influences interactions with proteins upstream of inflammasome assembly. Additionally, NLRP3 is reported to be recruited at the mitochondria and the endoplasmic reticulum after the inflammasome activation with MSU or nigericin [132]. Cardiolipin, a phospholipid from the outer mitochondrial membrane, interacts with the LRR domain, providing a docking site to localize activated NLRP3 at mitochondria. This interaction is crucial for inflammasome activation, indicating the involvement of the LRR domain in the activation process [133].

The evidence suggests that the LRR domain plays a crucial role in the activation process of the inflammasome. Cells expressing NLRP3 deletion mutant lacking the LRR domain were unable to secrete IL-1β in response to combined priming and activation signals. This indicates that the LRR domain is not only required for maintaining the endogenous stability of inactive oligomers but is also essential for NLRP3 inflammasome activation [120].

## 6. Typical and Atypical Cryopyrin-Associated Periodic Syndrome

Gain-of-function variants in NLRP3 are associated with CAPS, a rare monogenic autoinflammatory disease also named NLRP3-AID for NLRP3-associated autoinflammatory disease, which was first described in 2001 [64]. Most cases follow an autosomal dominant inheritance pattern, but somatic mutations are not rare and lead to sporadic acquired cases either in children, such as in the most severe form entitled NOMID/CINCA, or in adults [134]. Most pathogenic variants ultimately lead to overactivation of the NLRP3 inflammasome, resulting in the overproduction of IL-1β and, thus, cell pyroptosis. Normally, the NLRP3 inflammasome requires a priming signal and an activation signal to be active. The mechanism leading to the overactivation of the inflammasome depends on the location of the mutation, which can cause either a bypass of the priming signal, the activation signal, or both [135]. In the classical CAPS description, patients typically display episodes of fever and/or chills, pseudo-urticarial lesions triggered by environmental cold exposure, arthromyalgia +/− arthritis associated with neuroinflammatory features such as aseptic meningitis, sensorineural hearing loss, conjunctivitis, and uveitis [63]. While pseudo-urticarial lesions are a central symptom of classical CAPS and play a crucial role in clinical suspicion by clinicians, some patients may lack cutaneous involvement but can present with deafness, defining a subset classified as atypical CAPS [136,137].

Although the majority of variants are found in the NACHT domain, variants in the LRR domain are present in patients with atypical CAPS phenotypes associated with low or absence of cutaneous involvement and a higher risk of deafness [136,137]. This observation raises the question of a correlation between the location of the mutation and the atypical clinical phenotype with less frequent cutaneous features such as urticaria. We performed a literature review focusing on CAPS patients carrying *NLRP3* variants in the LRR domain. We found 45 patients reported; the *NLRP3* variants described were G755R (n = 5), G755A (n = 1), G809S (n = 1), K829T (n = 1), Y859C (n = 15), Y859H (n = 5), R918Q (n = 14), R918X (n = 2), L1016F (n = 1) (Table 1; Figure 8). The median age at symptom onset was 5 years, ranging from 0 to 32 years, and the median age at diagnosis was 34 years, ranging from 0.3 to 70 years. The median delay in diagnosis was 20 years, ranging from 0.3 to 49 years. Skin involvement with pseudo-urticarial lesions was found in 41.9% of patients (13/31), and hearing loss in 89.7% of cases (35/39). Atypical forms of CAPS were particularly prevalent in patients with the K829T, Y859C, Y859H, and R918Q variants of NLRP3, with 94.1% (32/34) of hearing loss and only 30.7% (8/26) of skin manifestations. In contrast, a significant cohort of CAPS patients with *NLRP3* mutated in the NACHT domain reported 42% of patients with hearing loss and 97% of patients with skin involvement [138]. Therefore, patients harboring *NLRP3* variants in the LRR domain tend to exhibit a higher incidence of hearing loss and lower prevalence of skin involvement compared to those with variants in the NACHT domain [137]. This observation supports a genotype-phenotype correlation depending on the domain carrying the variant: mutations in the NACHT domain would lead to typical CAPS presentation, while mutations in the LRR domain would be more likely to cause the atypical form of CAPS. These observations also suggest distinct pathophysiological mechanisms for NACHT-mutated and LRR-mutated *NLRP3* variants. Consistently, functional analysis of NLRP3 natural variants by Cosson et al. [135] showed that gain-of-function mutations in the LRR domain correspond to a new group of mutants activated by the activation signal independently of any priming. In opposition, most disease-causing variants from the NACHT domain are either constitutively activated or require only the priming signal to be active [135]. These observations suggest that atypical CAPS phenotype is preferentially caused by variations localized in the LRR domain of *NLRP3* that drive priming-independent but activation-dependent assembly of the inflammasome [135].

It remains unclear, however, how disease variants located in the LRR domain lead to abnormal inflammasome activation. At the molecular level, Q918, T755, and A755 mutations were described to enhance the interaction of the LRR domain with NEK7, potentially promoting NLRP3 activation [125]. In contrast, C859 and H859 were not predicted to impact NEK7-LRR interaction [136]. In cultured monocytes from patients with a Y859C genotype, the kinetics of IL-1β production demonstrate an increased production at an early time point and sustained production over time compared to cells with WT NLRP3 [136]. These observations might be explained by the role of Y859 in controlling NLRP3 degradation by phagolysosome, as shown in mice [98]. Alternatively, these mutations might disrupt the interaction between an acidic loop and the concave side of the LRR observed in the auto-inhibitory spheric cage complex [76], destabilizing this inactive complex.

In conclusion, our literature review on CAPS patients shows that patients displaying pathogenic *NLRP3* variants in the LRR domain present with more hearing loss and less frequent cutaneous urticaria lesions compared to patients with classical CAPS. Whether atypical forms of CAPS are more specifically related to *NLRP3* variants whose activation is independent of priming signal remains to be confirmed.

## 7. Conclusions/Discussion

NLRP3 is composed of three domains: the N-terminal PYD domain, the central NACHT domain, and the C-terminal LRR domain. Given the implication of NLRP3 in various diseases, including auto-inflammatory syndromes [55,56,57,64] and neuro-inflammatory diseases [40,41,42,59,60,61,62], deciphering the mechanism controlling NLRP3 inflammasome activation and degradation remains critical.

Compared to other proteins with LRR domains, the specific ligand for NLRP3 remains unidentified. Indeed, NLRP3 is responsive to a diverse array of stimuli, including microbial compounds, endogenous signals, and environmental triggers. Although evidence suggests that the LRR domain may be required to detect certain activator signals, no specific ligand-binding pocket has been discovered. It appears more likely that the known activators converge on cellular events, which lead to its activation. Specifically, NLRP3 activation is often associated with mechanisms involving K+ efflux, suggesting that this process could serve as the converging point and primary activator of NLRP3.

Pieces of evidence indicate that the LRR domain plays a critical role in regulating NLRP3. On one hand, interactions between LRR domains appear to maintain the integrity of the auto-inhibitory structure in steady-state oligomeric structures. On the other hand, PTMs accumulated within the LRR domains are documented to regulate both the activation and degradation of NLRP3.

Mutations in *NLRP3* associated with CAPS provide valuable insights into the functional roles of its LRR domain. While most mutations are found in the NACHT domain, those in the LRR domain lead to atypical symptoms, indicating a genotype-phenotype correlation. Unlike NACHT mutations, which typically activate NLRP3 by bypassing the activation signal, LRR mutations demonstrate a distinct activation profile that appears to be priming-independent and activation signal-dependent [135]. This unique activation pattern underscores the pivotal role of the LRR domain in the activation process of NLRP3. An intriguing aspect of these LRR domain mutations is the presentation of hearing loss as a primary symptom in affected patients. NLRP3 is expressed in immune cells in the cochlea [155], and K+ is known to play a critical role in auditory system functions [156]. Thus, the prevalent hearing loss observed in patients with LRR mutations might be explained by K+ efflux, which may act as the activation signal in the cochlea.

LRR-mutated patients are relatively rare, but their atypical clinical presentation suggests the actual incidence of these mutations may be underestimated. The delay in diagnosis might contribute to the evolution toward AA amyloidosis [157]. Early diagnosis is thus essential to initiate specific treatment as soon as possible to prevent outcomes such as deafness. In this context, IL-1β inhibitors have proven to be effective in reducing symptoms of the disease and in halting hearing loss progression [158,159,160].

The existence of several isoforms with unknown roles may impede cell- or tissue-specific expression and activity. It is widely accepted that among immune cells, mononuclear phagocytic cells express high levels of NLRP3 and are particularly responsive to activators that trigger inflammasome formation. Most studies are conducted in mouse models, human cell lines that have been genetically modified to overexpress NLRP3, or primary human monocytes. However, the exploration of NLRP3 regulation in other cell types remains limited, but future research could reveal cell-specific functions of the LRR domain. For example, NLRP3 is implicated in the formation of neutrophil extracellular traps [161,162], the differentiation of T cells, and the development of B cells [163], indicating a broader range of influence beyond inflammasome response. Further research into NLRP3 activity in different cell types might reveal cell-specific roles of the LRR domain.

## Figures and Tables

**Figure 1 cells-13-01365-f001:**
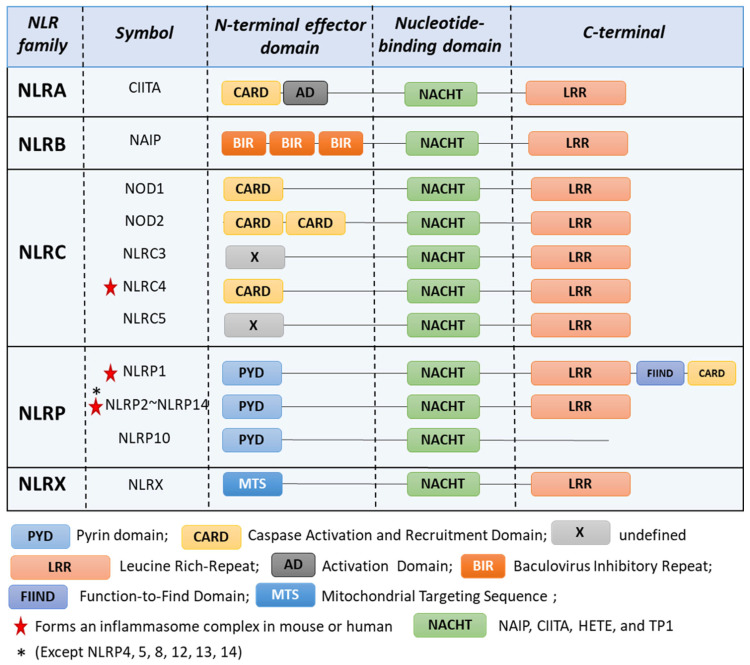
NOD-like receptors (NLRs) families. NLRs are characterized by a N-terminal, a central nucleotide-binding domain, and a C-terminal domain. NLRC4, NLRP1 [8], NLRP2 [9], NLRP3 [10], NLRP6 [11], NLRP7 [12], NLRP9 [13], NLRP10 [14], and NLRP11 [15] can form inflammasome.

**Figure 2 cells-13-01365-f002:**
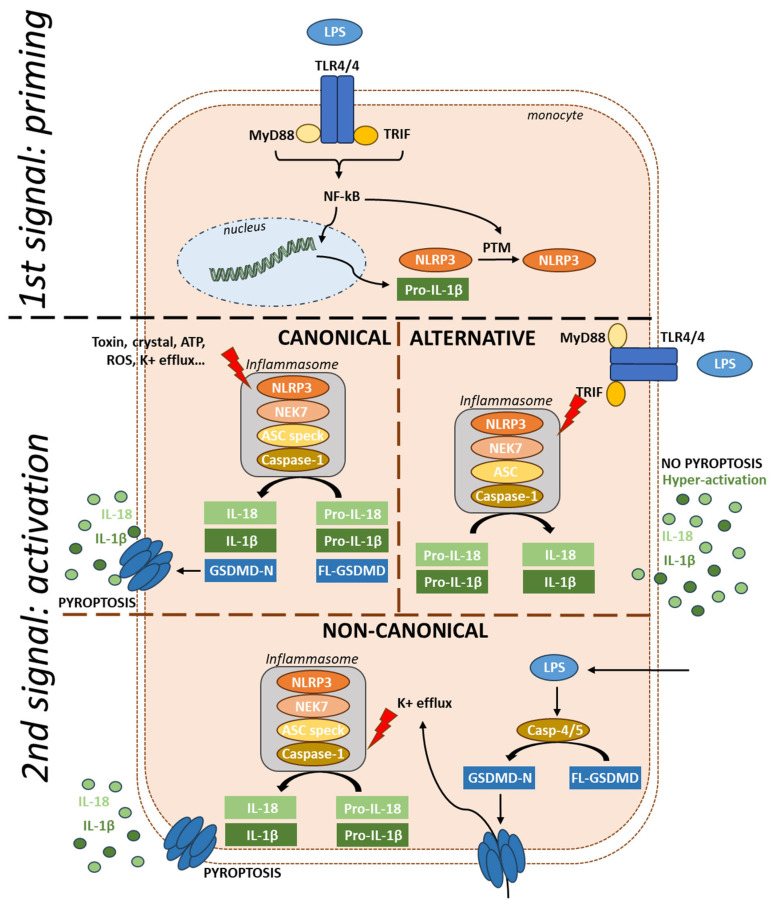
Activation mechanisms of NLRP3. The priming signal induces the production of NLRP3 and pro-IL-1β via TLR-NF-kB signaling, followed by a series of post-translational modifications. In canonical activation, the second signal, triggered by various activators, leads to the formation of the inflammasome with NEK7, ASC, and caspase-1. Once activated, the inflammasome cleaves pro-cytokines into their active forms and also cleaves gasdermin, leading to pyroptosis. In the alternative pathway, LPS via TLR4 signaling leads to the production of NLRP3 and pro-IL-1β. Dual activation through TRIF then activates the inflammasome, resulting in hyperactivation without pyroptosis and no ASC speck formation. In the non-canonical pathway, intracellular LPS serves as the second signal, activating caspases 4 and 5, which cleave gasdermin, leading to pyroptosis. Pores in the membrane cause K+ efflux, triggering NLRP3 inflammasome formation. FL-GSDMD: Full-length gasdermin; GSDMD-N: Gasdermin-N-terminal; NF-kB: Nuclear Factor kappa-light-chain-enhancer of activated B cells.

**Figure 3 cells-13-01365-f003:**
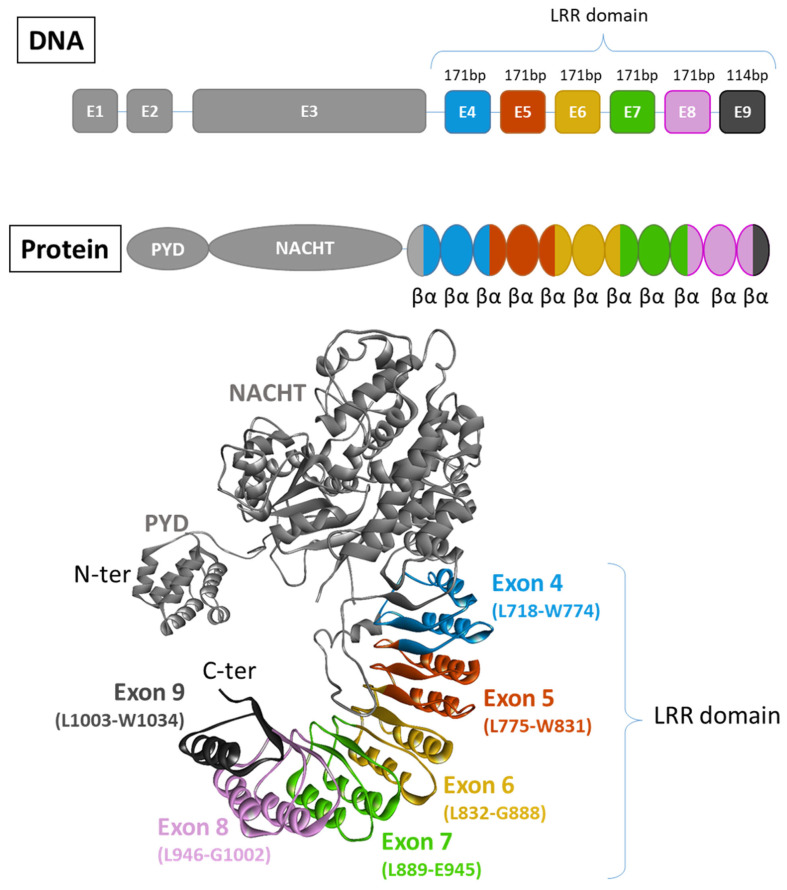
Exon organisation of NLRP3. There are two recognized versions of the NLRP3 gene nomenclature: one version includes 10 exons [72], while the other comprises a total of 9 exons [73,74]. In this latter version, 6 of them contribute to the formation of the LRR domain. Each exon code for 2 completes LRR motifs composed of 2 α-helix and 2 β-sheets (PDB model: 7PZC [76]).

**Figure 4 cells-13-01365-f004:**
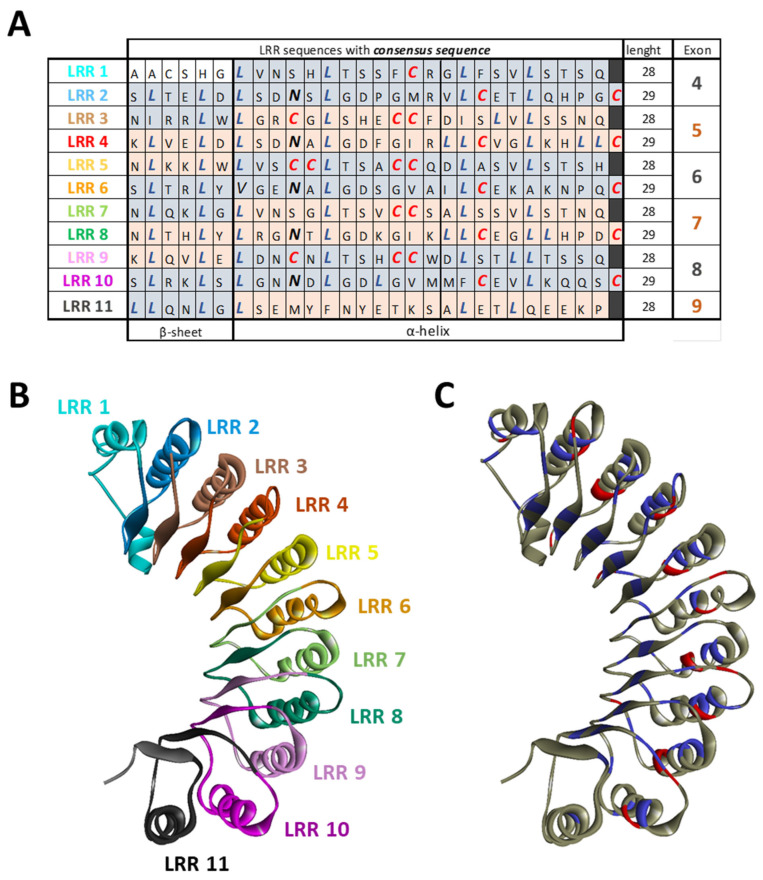
LRR organization of NLRP3. (**A**) Each exon code for 2 completes LRR motifs alternating 28 or 29 amino acids. NLRP3 is composed of 11 LRRs that are characterized by a high leucine (blue) and cysteine (red) content and a consensus sequence. (**B**) The structure of the LRR domain with each LLR repeat is highlighted. (**C**) Structure of the LRR domain with leucine in blue and cysteine in red. (PDB model: 7PZC).

**Figure 5 cells-13-01365-f005:**
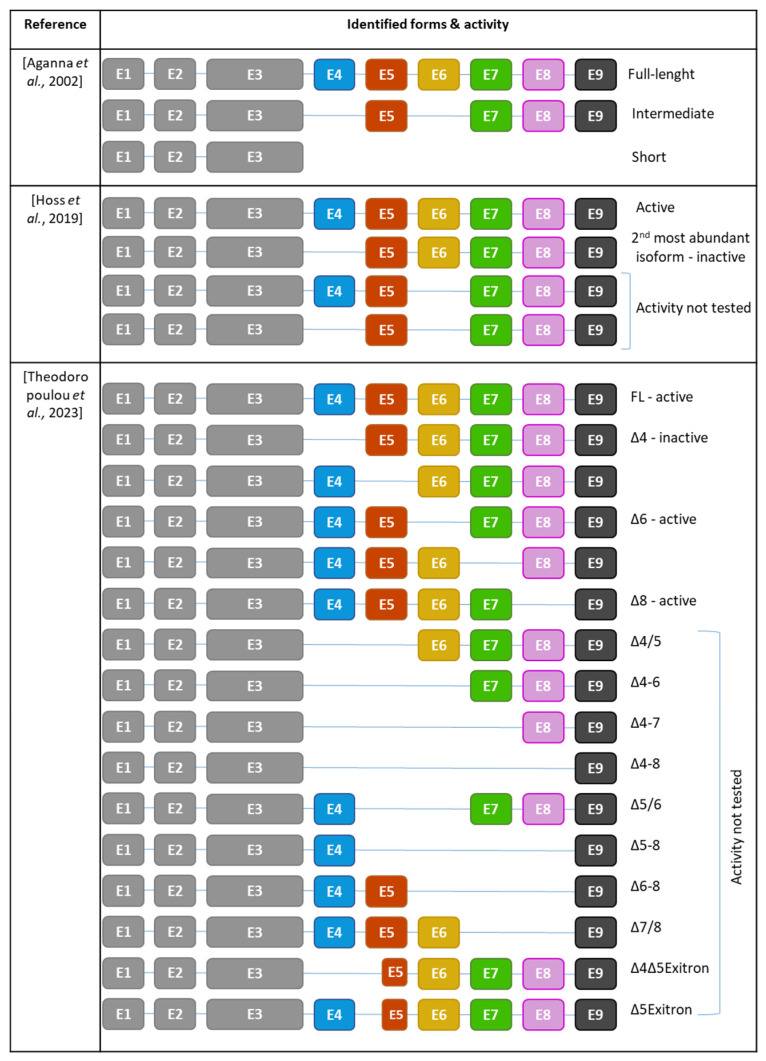
Repertoire of NLRP3 isoforms identified and their activities. E: exon [72,73,74].

**Figure 6 cells-13-01365-f006:**
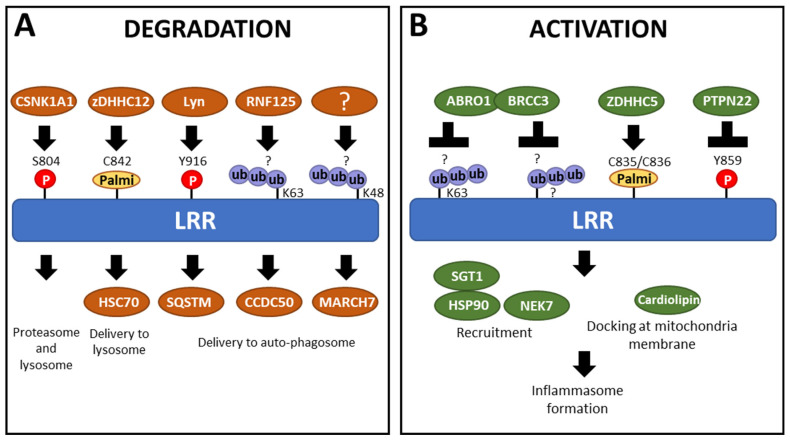
Partners alter the PTM status of the LRR domain, participating in either (**A**) degradation or (**B**) activation of NLRP3. Ub = ubiquitination; Palmi = palmitoylation; P = phosphorylation; ? = amino acids carrying ubiquitination are unknown.

**Figure 7 cells-13-01365-f007:**
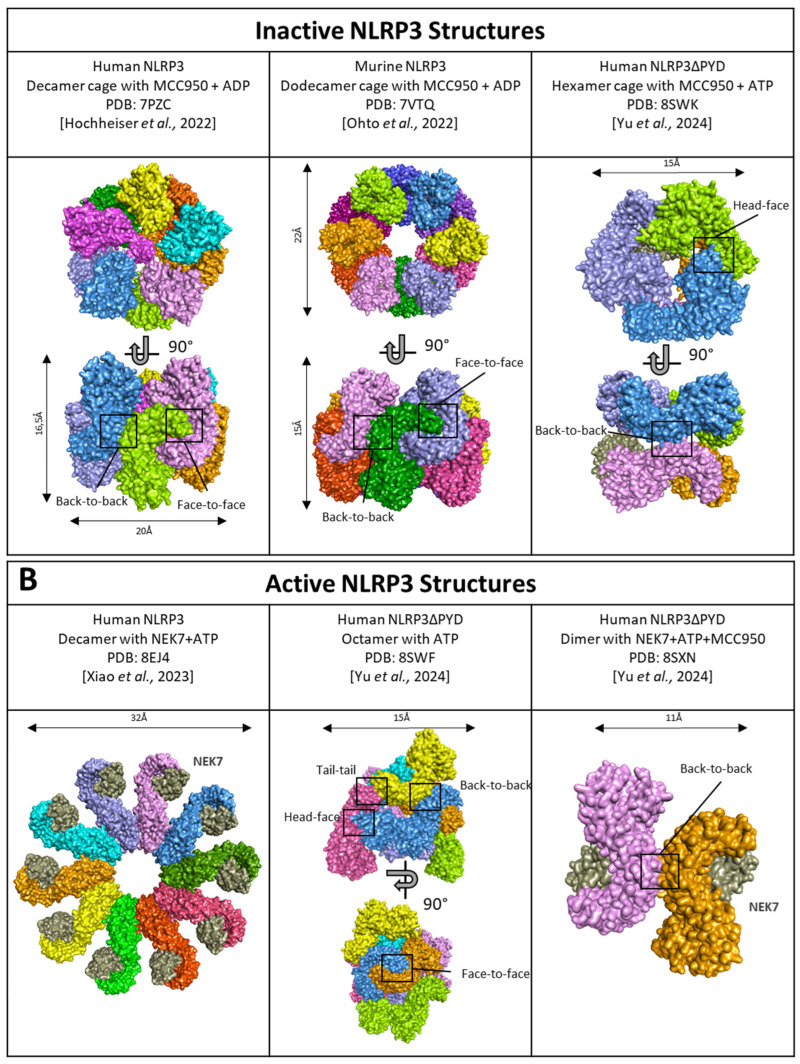
Structure of (**A**) inactive NLRP3 cages and (**B**) active NLRP3. NLRP3 monomers are represented in several colors. NEK7, a partner of activation is in grey. Black boxes highlight the interaction surfaces. PDB ID: 7PZC [76]; 7VTQ [117]; 8EJ4 [119]; 8SWF/8SXN/8SWK [116].

**Figure 8 cells-13-01365-f008:**
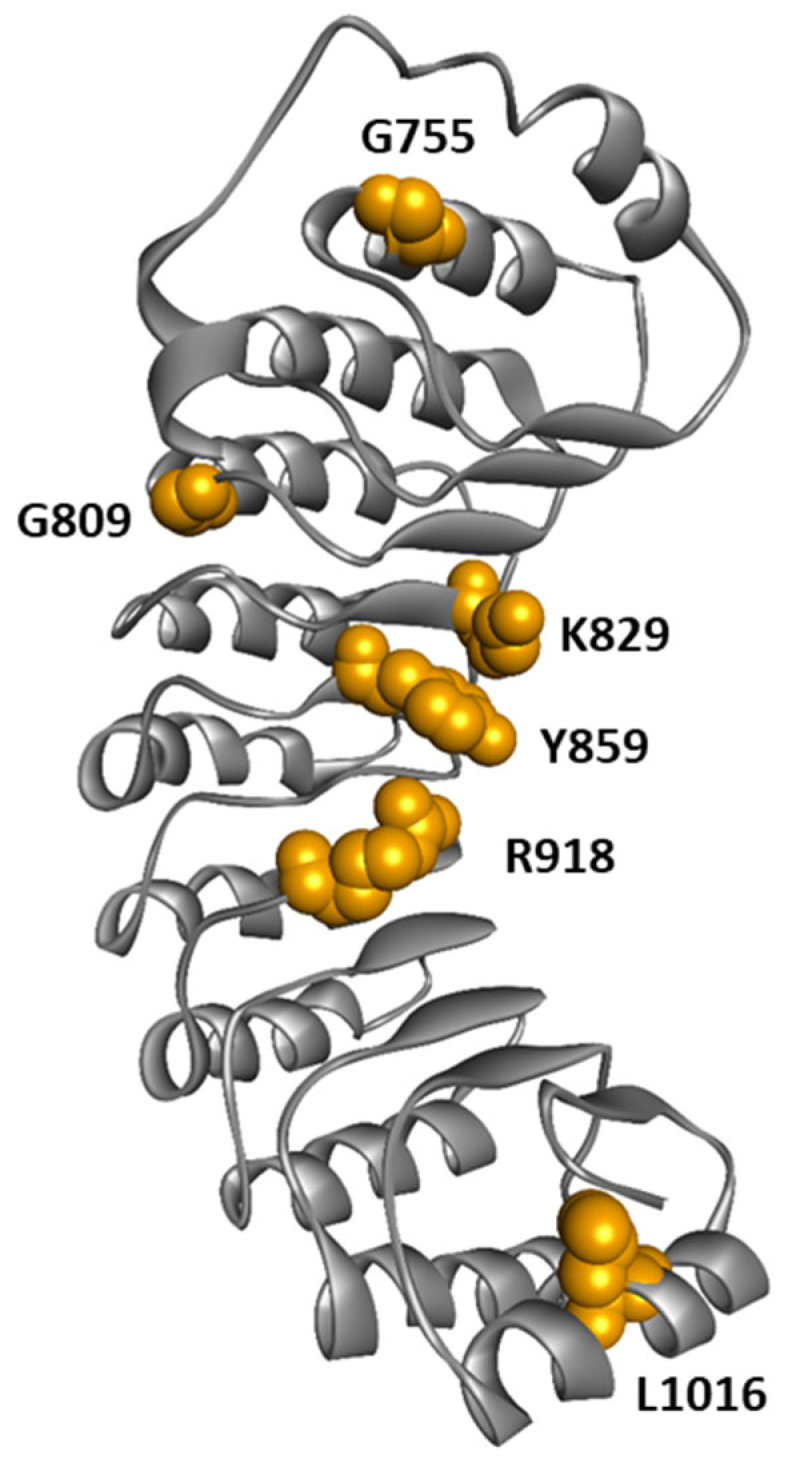
Structure of NLRP3-LRR highlighting amino acids mutated in CAPS patients.

**Table 1 cells-13-01365-t001:** A literature review focusing on pathogenic LRR variants was performed using the PubMed database. A total of 1113 articles were retrieved. After removing duplicates and excluding articles describing patients with mutations in the NACHT domain, those without genetic analysis, those describing previously reported patients, and those with other pathogenic mutations explaining the symptoms, a total of 18 articles describing 45 patients were included in the final analysis.

Article	Genotype	Sex	Age at Onset (Years Old)	Age at Diagnosis (Years Old)	Fever	Urticaria	Arthromyalgia and/or Arthritis	Meningitis	Uveitis and/or Papillary Edema	Hearing Loss *	Hearing Aids	Amyloidosis AA
[139]	p.G755R	M	Birth	N/D	+	+	+	N/D	+	N/D	N/D	+
[140]	p.G755R	N/D	Birth	N/D	+	+	+	+	+	1, Mild	N/D	N/D
[141]	p.G755R	N/D	N/D	N/D	N/D	N/D	N/D	N/D	N/D	N/D	N/D	N/D
[142]	p.G755R	F	N/D	3	N/D	+	+	+	N/D	N/D	N/D	N/D
[143]	p.G755R	F	Birth	0.3	+	+	−	+	N/D	−	N/D	N/D
[144]	p.G755A	N/D	1	N/D	N/D	N/D	N/D	N/D	N/D	N/D	N/D	N/D
[145]	p.G809S Ϯ	M	<1	N/D		+	+	+	N/D	−	−	N/D
[146]	p.K829T	M	2	20	+	−	−	N/D	−	+	N/D	N/D
[147]	p.Y859C	M	2	16	+	−	+	−	+	+, Moderate	+	N/D
[148]	p.Y859C	M	12	N/D	N/D	N/D	N/D	N/D	N/D	N/D	N/D	N/D
[149]	p.Y859C	F	18	N/D	−	−	−	+	+	+, Severe	+	−
	p.Y859C	M	N/D	N/D	−	−	−	−	+	+, Moderate	N/D	−
[150]	p.Y859C	F	4	7	+	−	−	N/D	+	+, Moderate	N/D	N/D
[151]	p.Y859C	F	Childhood	38	+	+	+	N/D	N/D	+	N/D	N/D
[136]	p.Y859C	N/D	32	54	N/D	−	+	−	−	+	N/D	−
	p.Y859C	N/D	3	30	N/D	−	+	+	+	+	N/D	−
	p.Y859C	N/D	3	38	N/D	−		+	+	+	N/D	−
	p.Y859C	N/D	20	38	N/D	−	+	−	+	+	N/D	−
	p.Y859C	N/D	10	44	N/D	−	+	+	−	+	N/D	−
	p.Y859C	N/D	13	50	N/D	−	+	+	−	+	N/D	−
	p.Y859C	N/D	3	7	N/D	+	−	−	+	+	N/D	−
	p.Y859C	N/D	2	12	N/D	+	+	+	+	+	N/D	−
	p.Y859C	N/D		34	N/D	−	+	−	+	+	N/D	−
	p.Y859H	N/D		16	N/D	+	−	−	−	+	N/D	−
	p.Y859H	N/D	11	16	N/D	−	+	−	+	+	N/D	−
	p.Y859H	N/D	8	47	N/D	−	+	+	−	+	N/D	−
	p.Y859H	N/D	6	21	N/D	−	+	−	−	+	N/D	−
	p.Y859H	N/D		49	N/D	+	+	−	−	+	N/D	−
[125]	p.R918Q	F	9	N/D	+	−	+	N/D	N/D	+, Mild	−	N/D
[152,153]	p.R918Q	F	30	59	−	−	+	N/D	N/D	+, Moderate	N/D	N/D
	p.R918Q	F	N/D	69	−	−	−	N/D	N/D	+, Moderate	N/D	N/D
	p.R918Q	F	N/D	32	N/D	N/D	N/D	N/D	N/D	−	−	N/D
	p.R918Q	M	N/D	35	+	+	+	N/D	N/D	+, Moderate	N/D	N/D
	p.R918Q	M	N/D	13	+	N/D	N/D	N/D	N/D	+	N/D	N/D
	p.R918Q	F	N/D	10	+	+	N/D	N/D	N/D	+	N/D	N/D
	p.R918Q	M	N/D	6	+	+	N/D	N/D	N/D	−	−	N/D
	p.R918Q	F	N/D	70	N/D	N/D	N/D	N/D	+	+, Severe	N/D	N/D
	p.R918Q	M	N/D	60	N/D	N/D	N/D	N/D	+	+, Severe	N/D	N/D
	p.R918Q	D	N/D	50	N/D	N/D	N/D	N/D	+	+, Severe	N/D	N/D
	p.R918Q	D	N/D	40	N/D	N/D	N/D	N/D	+	+, Moderate	N/D	N/D
	p.R918Q	M	N/D	30	N/D	N/D	N/D	N/D	+	+, Moderate	N/D	N/D
	p.R918Q	M	N/D	40	N/D	N/D	N/D	N/D	+	+, Moderate	N/D	N/D
[154]	p.R918X	F	N/D	N/D	N/D	N/D	N/D	N/D	N/D	+, Severe	+	N/D
	p.R918X	F	1.5	N/D	N/D	N/D	N/D	N/D	N/D	+, Mild	N/D	N/D
[138]	p.L1016F #	N/D	N/D	N/D	N/D	N/D	N/D	N/D	N/D	N/D	N/D	N/D

* Severity of hearing loss: mild if loss <40 dB, moderate if between 40 and 70 dB, severe if >70 dB or defined as so by the authors. Ϯ Heterozygous patient with a mutation in the NACHT domain (E688K). # In vitro analysis showed no gain-of-function of this mutation [135].

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
