# Peer review of "Implication of the LRR Domain in the Regulation and Activation of the NLRP3 Inflammasome"

_cells, 2024, doi:10.3390/cells13161365_

Round 1

Reviewer 1 Report

Comments and Suggestions for Authors

The manuscript ‚Implication of the LRR domain in the regulation and activation of the NLRP3 inflammasome’ by Cescato et al. provides a very interesting, comprehensive and well-written overview on the role of the leucine-rich repeats found in the most studied inflammasome sensor protein Nlrp3. The authors put a lot of effort in providing an up-to-date review including extensive discussion of structural properties of the domain, various reported and tested isoforms of Nlrp3, post-translational modifications, and clinical relevance of the LRR domain. While the review article is well-suited for publication, the authors need to address a few points before publication:

-          While the article clearly focusses on Nlrp3, it would be very interesting to mention Nlrp10 in the main text, as it is the only NLRP without LRR. Recent work by Próchnicki et al. and Zheng D et al., both 2023 Nat. Immunol., highlighted this yet less well studied NLR and it might be relevant to discuss its potential role as sensor or even inflammasome regulator lacking the LRR domain.

-          The authors nicely summarize and clarify the confusion regarding the different aa annotations found for NLRP3 in the literature. However, while I understand the concept, I would be hesitant calling the annotations US and European systems, as the use of each system is certainly not restricted to continents and no clear consensus in terms of nomenclature has been reached here yet to my knowledge. I suggest the authors in addition refer to the initial papers introducing the respective annotation system, e.g. ‘In the US system, according to Hoffman et al., …’ and similar for the European system).

-          In Figure 3, the authors comprehensively depict both LRR structure and sequences. While in the sequence table each found LRR is stacked to appreciate the conserved leucine position in particular, other repetitive characteristics alternating in every other LRR are not highlighted. In particular, the prominent repetitions of cysteines are neither appreciated in this depiction nor in the main text. Therefore, the authors need to implement a section discussing the potential role of the LRR cysteines as those residues might harbor a function in ROS sensing. The authors could also refer to a review article by Neuwirt et al., Curr Opin Biotechnol. 2021, which already discussed that NLRP3 has an unusually high content of the redox-sensitive amino acid cysteine and provided both sequence alignments and structural details on the positions.

-          The authors write ‘The use of ROS scavengers has shown no effect on inflammasome activation, suggesting that ROS are not directly implicated (29).’ followed by ‘However, imiquimod, a small molecule known to be a ligand of TLR7, activates NLRP3 in a K+-independent but ROS-dependent manner (42).’ in one paragraph which feels contradictory. Please revise.

-          In Figure 6 the authors provide an appealing depiction of Nlrp3 structures. For 6B I recommend to select a unique color for the structure of NEK7, as readers might get confused by the same light green used in other panels of the figure.

-          Table 1 is a complex table over 3 pages. Would it be possible to repeat the header on every page? Separation or shading of the columns might also be helpful. Also, the legend to the table seems to start with a truncated sentence.

-          In the references the publication months are given in French.

-          Check the sentence: This review aims to provide a comprehensive overview of current knowledge and recent discoveries regarding the involvement of LRR NLRP3 domain in modulating NLRP3 inflammasome activation.

-          Check the sentence: Considering that the physiological intracellular concentration of K+ is too high to active NLRP3 and that inhibiting K+ efflux prevents NLRP3 activation in both murine and human macrophages (75), it suggests that K+ efflux may be the critical convergence point for NLRP3 activation (29).

- Check the sentence: NLRP3 mutant lacking the LRR domain was unable to produce IL-1β following both the priming and activation steps.

Author Response

We thank reviewer 1 for all his comments, helping our manuscript to gain clarity.

Best regards.

Reviewer 2 Report

Comments and Suggestions for Authors

Cescato M. et al, in the Review “Implication of the LRR domain in the regulation and activation of the NLRP3 inflammasome” clearly describe the structure and function of LRR domain highlighting its involvement in NLRP3 inflammasome activation and the impact of its mutations in pathological conditions.

The Review is well written, organized and focused. The Review is well referenced, and the Figures are fine prepared.

Minor:

1. I suggest improving the description of the different NLRP3 activation pathways (canonical, non-canonical, alternative) and add an explanatory figure of these pathways, to better focus the involvement of LRR in the regulation and activation of NLRP3 inflammasome.

2. The Authors said: “Furthermore, gain-of-function heterozygous variants in NLRP3 are associated with a rare monogenic auto-inflammatory disease called Cryopyrin-Associated Periodic Syndrome (CAPS) and characterized by systemic inflammation and various clinical features including mainly urticaria, arthromyalgia, buccal aphtous, and neuroinflammation with eye, hear and mengingitis involvement” Please add the specific bibliography.

3. Please check the title of Table 1.

4. Please check the fonts that appears to be different within the Review in all sections and notice that some figures are highlighted.

Author Response

We thank reviewer 2 for his comments, helping us to improve our manuscript.

Best regards.

Reviewer 3 Report

Comments and Suggestions for Authors

This is a very readable and informative review concerning the complex activation and interactions of NLRP3 and other NLR families, with a focus on the LRR domain. The role of PTM in activation/deactivation/regulation of NLRP3 is discussed very well. The Figures are of outstanding quality. Table 1 / Figure 7 is therapeutically very interesting, giving a summary of patients with CAPS, with their mutations in LLR. Overall, a very useful review for experts in this very active field, as well as scientists with an interest in inflammation.

Comments on the Quality of English Language

English is very good.

Minor suggestions:

p9, line 5: 'active' should be 'activate'

Font and size needs attention in  abstract and introduction.

Author Response

Comment: p9, line 5: 'active' should be 'activate'. Font and size needs attention in abstract and introduction.

RESPONSE:  We thank reviewer 3 for his comment. The spelling mistake was modified accordingly and the manuscript fonts are harmonized.